# Imputing Incomplete Building Attribute Data Using Expert-Guided Variational Autoencoders (EGVAE).

## Abstract

Human activity is organized by the physical infrastructure such as roads and buildings, making their characterization essential for applications ranging from disaster preparedness and national security to real estate analytics and public resource allocation. Despite the availability of datasets detailing building attributes many of these are incomplete, particularly in data scarce regions, limiting their utility in critical decision making tasks. We propose a deep learning approach for imputing missing building attributes by learning from sparse to no observed data, expert knowledge, and spatial correlations among buildings. Our model is based on a Vector Quantised-Variational AutoEncoder (VQ-VAE) architecture with a graph neural network (GNN) encoder that captures spatial dependencies, while an additional KL-divergence based loss term incorporates expert-informed priors. By jointly leveraging observed data and expert-informed priors, the model learns latent representations that enable imputing missing data for attributes with little or no training data. Experimental results on real-world datasets demonstrate the robustness and effectiveness of our proposed method.

## 1 Introduction

Buildings, roads and among other infrastructure is how humans organize there daily lives. Understanding key building attributes, such as, building use type, floor counts, construction material, and foundation is critical for application in disaster management, national security, real estate development, and equitable resource allocation. Accurate building attribution can help inform policy makers with decision making at local and national levels.

Several publicly available datasets aim to capture this information, including OpenStreetMap (OSM) OpenStreetMap contributors (2017), Microsoft's US Building Footprints Microsoft (2018), and FEMA's USA Structures dataset Yang et al. (2024). While these datasets provide valuable coverage, they are often incomplete especially in developing regions or rural areas.

Manual surveys and remote sensing approaches can partially address these gaps, but they are costly, time-consuming, and difficult to scale globally. To bridge this gap, there is a growing need for scalable, automated approaches to impute missing building attribute data using machine learning methods that can generalize across geographies and data regimes. We posit that building attributes arise from an underlying structural latent representation, which can be learned and exploited for imputation.

In this work, we propose a deep learning approach to address the challenge of incomplete building attribute datasets. The method is built on a Vector Quantised-Variational AutoEncoder (VQ-VAE) van den Oord et al. (2018); Nazábal et al. (2018) framework, with a graph neural network (GNN) Zhou et al. (2021) encoder that learns spatial relationships between neighbouring buildings. These spatial cues, such as proximity and shared neighborhood characteristics often carry valuable implicit information Besag & Kooperberg (1995) about building types and attributes.

Moreover, we enhance the VQ-VAE with an expert-elicitation guided training, introduced through a KL-divergence-based loss term. By combining learned data representations with expert-guided constraints, the model produces imputations that are aligned with domain knowledge.

We validate the proposed approach using real-world data from sources such as OpenStreetMaps (OSM) and USA Structures. The results show that the proposed model outperforms traditional imputation methods, particularly in data-poor settings where standard approaches struggle.

Our key contributions are as follows:

- We introduce the first application of VAEs for imputing incomplete building data, providing a fast and scalable solution to a long standing challenge.
- We exploit structural distinctions in building types by shaping the latent space with discrete building representations via vector quantization.
- We leverage aggregate-level expert priors into latent space, enabling informed imputations in regions with little or no observed data.
- We enable testing of domain assumptions about the built environment by encoding them explicitly in a directed acyclic graph.

## 2 METHOD

We propose a **Expert-Guided Variational Autoencoder (EGVAE)** with a graph neural network encoder, a vector quantization bottleneck, and a multi-head decoder. We also assume a directed acyclic graph (DAG) that encodes relationships between building attributes. The model is trained to minimize reconstruction error while respecting a prior distribution defined by expert elicitations that are represented as conditional probability tables (CPTs). The CPTs describe numerical probabilities associated with each dependency in the DAG.

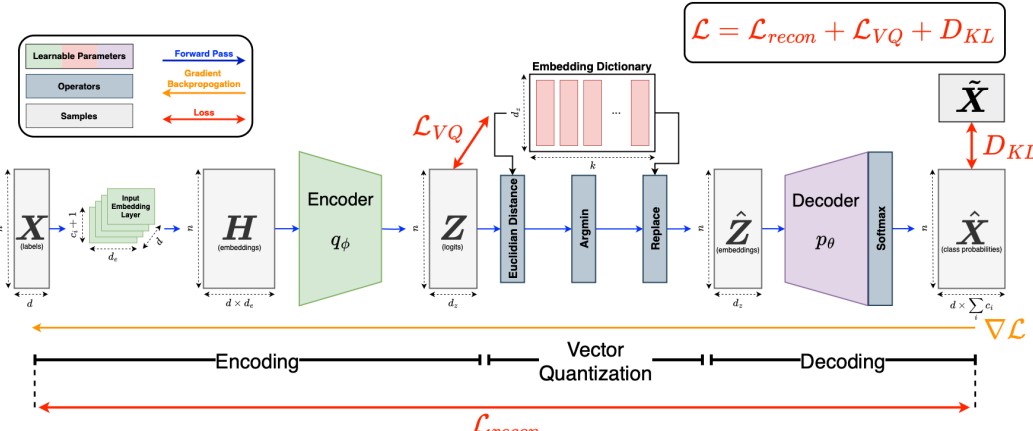

Figure 1: The forward pass involves processing input data consisting of $n$ buildings, each characterized by $d$ categorical attributes, with each attribute having $c_i \in (c_0, \cdots, c_{d-1})$ possible categories. Each attribute in the input data is embedded into a space of dimension $d_e$, with an additional category to account for missing values. These individual embeddings are concatenated into an input data matrix $\boldsymbol{H}$, which is then mapped to a latent space of dimension $d_z$ by the encoder $q_\phi$. The resulting latent representations $\boldsymbol{Z}$ are quantized to one of $k$ prototype vectors, which represent the embeddings in the latent space. Finally, the quantized embeddings $\hat{\boldsymbol{Z}}$ are projected into a probability vector space $\hat{\boldsymbol{X}}$ using the decoder $p_\theta$ with a softmax normalization.

### 2.1 PROBLEM SETUP

Let $\boldsymbol{X} \in \{-1, 0, \ldots, c_i-1\}^{n \times d}$ denote a categorical tabular dataset with $n$ buildings and $d$ building attributes (features), where each feature $\boldsymbol{X}_i$ with $c_i$ categories takes values from $\{-1, 0, \ldots, c_i - 1\}$ and $-1$ represents missing data. Dependencies between features are modeled using a directed acyclic graph $\mathcal{D} = (V, E)$, with nodes corresponding to features and edges indicating conditional relationships between them. While another directed graph $\mathcal{G} = (B, P)$ represents buildings $B$ and

edges $P$ derived using adjacency matrix $\boldsymbol{A}$ corresponding to buildings in the 20 meter proximity of each building.

## 2.2 MODEL ARCHITECTURE

### 2.2.1 ENCODER

Each feature $\boldsymbol{X}_i$ is embedded using a learnable embedding matrix $\boldsymbol{E}_i \in \mathbb{R}^{(c_i+1) \times d_e}$, where the last row handles the missing value indicator. For a batch of instances, the full input embedding is:

$$\boldsymbol{H} = \text{concat}\left([\boldsymbol{E}_i(\boldsymbol{X}_i)]_{i=1}^d\right)$$

We apply a three-layer Graph Convolutional Network (GCN) Kipf & Welling (2017) with ReLU activations and 35% MC Dropouts to encode these embeddings, using the adjacency matrix $\boldsymbol{A}$ of $\mathcal{G}$:

$$\boldsymbol{Z}^{(1)} = \text{MCDropout}(\sigma(\text{GCNConv}_1(\boldsymbol{H}, \boldsymbol{A}))) \tag{1}$$

$$\boldsymbol{Z}^{(2)} = \text{MCDropout}(\sigma(\text{GCNConv}_1(\boldsymbol{Z}^{(1)}, \boldsymbol{A}))) \tag{2}$$

$$\boldsymbol{Z} = \text{GCNConv}_2(\boldsymbol{Z}^{(2)}, \boldsymbol{A}) \tag{3}$$

where $\sigma$ denotes the ReLU activation function and $\boldsymbol{Z} \in \mathbb{R}^{n \times d_z}$ is the latent representation.

### 2.2.2 VECTOR QUANTIZATION

To enforce discrete latent variables, we use vector quantization with a learnable codebook $\mathcal{E} = \{\boldsymbol{e}_j\}_{j=1}^k \subset \mathbb{R}^{d_z}$. Each latent vector $\boldsymbol{z}_i$ is quantized to its nearest codebook vector:

$$\hat{\boldsymbol{z}}_i = \boldsymbol{e}_l, \quad \text{where } l = \arg\min_j \|\boldsymbol{z}_i - \boldsymbol{e}_j\|^2$$

$$\hat{\boldsymbol{Z}} = \text{concat}([\hat{\boldsymbol{z}}_i]_{i=1}^n)$$

### 2.2.3 DECODER

The quantized representations $\hat{\boldsymbol{Z}}$ are passed through a shared hidden layer and then to separate output heads for each feature:

$$\boldsymbol{h} = \text{ReLU}(\boldsymbol{W}\hat{\boldsymbol{z}} + \boldsymbol{b}) \tag{4}$$

$$\hat{\boldsymbol{x}}_i = \text{Softmax}(\boldsymbol{W}_i\boldsymbol{h} + \boldsymbol{b}_i) \tag{5}$$

where $\boldsymbol{W}$'s, $\boldsymbol{b}$'s are learnable weights and biases, and $\hat{\boldsymbol{x}}_i \in \mathbb{R}^{c_i}$ is the predicted categorical distribution for the $i$ feature.

## 2.3 LOSS FUNCTION

The total training loss combines the reconstruction loss, vector quantization loss and CPT regularization:

$$\mathcal{L}_{\text{total}} = \mathcal{L}_{\text{recon}} + \mathcal{L}_{\text{CPT}} + \mathcal{L}_{\text{VQ}}$$

each described below.

## 2.4 RECONSTRUCTION LOSS

To account for missing data, we compute the reconstruction loss only on observed entries using cross-entropy:

$$\mathcal{L}_{\text{recon}} = \sum_{i=1}^d \frac{1}{|\mathcal{O}_i|} \sum_{j \in \mathcal{O}_i} \text{CrossEntropy}(\hat{\boldsymbol{x}}_i^{(j)}, \boldsymbol{x}_i^{(j)})$$

where $\mathcal{O}_i = \{j \mid \boldsymbol{x}_i^{(j)} \neq -1\}$ denotes the set of observed indices for feature $i$.

### 2.4.1 VECTOR QUANTIZATION LOSS

We follow the formulation of van den Oord et al. (2018) and define the vector quantization loss as:

$$\mathcal{L}_{\text{VQ}} = \|\text{sg}[\boldsymbol{z}] - \hat{\boldsymbol{z}}\|_2^2 + \beta \|\boldsymbol{z} - \text{sg}[\hat{\boldsymbol{z}}]\|_2^2$$

Here, the operator $\text{sg}[\cdot]$ denotes the *stop-gradient* operation, which treats its argument as a constant during backpropagation. Specifically, $\text{sg}[\boldsymbol{z}]$ blocks gradients from flowing into the encoder, allowing the codebook to be updated independently, while $\text{sg}[\hat{\boldsymbol{z}}]$ ensures the encoder receives gradients to align its output with the quantized vectors.

### 2.4.2 CPT-BASED REGULARIZATION

To incorporate prior knowledge, we regularize the model using conditional probability tables (CPTs) associated with a Bayesian network structure $\mathcal{D}$. For each variable $\boldsymbol{X}_i$, let $\mathbf{Pa}(\boldsymbol{X}_i)$ be the set of its parent variables. The CPT defines a conditional distribution $\boldsymbol{P}(\boldsymbol{X}_i \mid \mathbf{Pa}(\boldsymbol{X}_i))$.

Let $\hat{\boldsymbol{x}}_i$ be the predicted distribution and $\tilde{\boldsymbol{x}}_i$ be the target CPT-derived distribution. We define the CPT loss as the KL-divergence between the two:

$$\mathcal{L}_{\text{CPT}} = \sum_{i=1}^{d} D_{KL}(\hat{\boldsymbol{x}}_i \,\|\, \tilde{\boldsymbol{x}}_i)$$

If $\boldsymbol{X}_i$ has no parents, $\tilde{\boldsymbol{x}}_i$ is the marginal distribution; otherwise, it is selected based on the predicted parent configuration.

## 3 EXPERIMENTS

### 3.1 EXPERIMENTAL SETUP

#### 3.1.1 DATASETS

Table 1: Dataset Description: Percentage of Missing Values and Expert Priors Available.

| Attribute | Nashville (N=18,390) | | Amman (N=428,494) | | Mexico City (N=3,194,984) | |
|---|---|---|---|---|---|---|
| | % Missing | Priors | % Missing | Priors | % Missing | Priors |
| purpose | 0.0 | ✓ | 33.04 | ✓ | 7.25 | ✓ |
| area | 15.51 | ✓ | 0.00 | ✗ | 0.00 | ✗ |
| facade | 44.80 | ✓ | 100.00 | ✓ | 100.00 | ✗ |
| floors | 0.14 | ✓ | 0.01 | ✓ | 99.94 | ✓ |
| roof shape | 100.00 | ✗ | 100.00 | ✓ | 100.00 | ✗ |
| roof material | 46.18 | ✓ | 100.00 | ✗ | 100.00 | ✗ |
| material | 100.00 | ✓ | 100.00 | ✓ | 100.00 | ✓ |
| foundation | 100.00 | ✗ | 100.00 | ✗ | 100.00 | ✗ |
| llrs | 100.00 | ✓ | 100.00 | ✓ | 100.00 | ✓ |

We evaluate the proposed model on three separate real-world tabular datasets with $d = 9$ building attributes for Amman, Jordan, and Mexico City. Due to the lack of publicly available datasets that jointly provide building footprints, building attributes, and expert priors, we rely on internally curated datasets for evaluation. All three datasets have a considerable amount of missing building attributes with expert prior information on some attributes (Table 1). Additionally, we assume the same graph structure $\mathcal{D}$ describing the relationships between building attributes for all three datasets (Figure 2).

Additionally, we also report results on a synthetic dataset with 4 building attributes - area, height, material and purpose. We first define the DAG (see Figure 3) and the associated CPTs. The dataset is generated by drawing ($N = 5000$) samples from the CPTs.

### 3.1.2 CONDITIONAL PROBABILITY TABLES (CPTS)

Many building attributes such as roof shape, floor counts, external facade are not independent and therefore cannot be described by marginal distributions alone. They exhibit structured, hierarchical relationships for example, the likelihood of a building being classified as tall is strongly dependent on area occupied by the building footprint and the use type of the building, while the lateral load resistance system (LLRS) of a building is conditional on both the floor count of the building and the primary construction material (see Figure 2).

Conditional probability tables (CPTs) provide a natural mechanism for representing such dependencies. Formally, a CPT specifies the distribution of a categorical attribute $\mathbf{X_i}$ conditioned on a set of parents attributes $\mathbf{Pa}(\boldsymbol{X}_i)$. Each row of the table encodes

$$P(\boldsymbol{X}_i \mid \mathbf{Pa}(\boldsymbol{X}_i) = \boldsymbol{p}), \qquad (6)$$

where $p$ denotes a particular configuration of parent attribute values. The full set of CPTs defines a directed acyclic graph over building attributes, ensuring the joint distribution factors as

$$P(\boldsymbol{X}_1, \cdots, \boldsymbol{X}_n) = \prod_{i=1}^{d} P(\boldsymbol{X}_i | \mathbf{Pa}(\boldsymbol{X}_i)) \qquad (7)$$

Figure 2: Directed acyclic graph $G$

This representation enables explicit incorporation of domain knowledge. Structural engineers and urban planners can specify plausible or implausible attribute combinations directly in the tables, while also assigning relative probabilites to reflect common patterns observed in the built environment. Predefined structures such as Global Exposure Model Yepes-Estrada et al. (2023) provide a global coverage on some of the building attributes that are leveraged in this study. The CPT framework thus provides two critical functions in our modeling approach, First, it ensures that imputed attributes remain internally consistent, respecting the conditional structures encoded by experts. Second, it allows probabilistic models to integrate heterogeneous sources of information, while prior CPT specifications act as regularizers in sparse-data contexts. This hybrid approach is particularly valuable in regions where detailed building attribute labels are unavailable but expert knowledge of local construction practices is strong.

### 3.1.3 IMPLEMENTATION DETAILS

We use the following hyperparameters for all models

- **Embedding Dimension:** $d_e = 16$
- **Hidden Dimension:** $d_h = 64$
- **Latent Dimension:** $d_z = 32$
- **Codebook Size:** $k = 64$
- **Commitment Cost:** $\beta = 0.25$

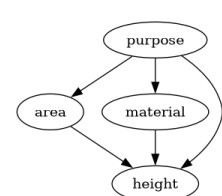

Figure 3: DAG for synthetic data

We use the Adam optimizer with a learning rate of $1e^{-3}$. The model is trained for 500 epochs for synthetic data on Apple M2 processor. Amman, Nashville and Mexico city data is trained for 500, 500 and 1000 epochs respectively on a 64-bit AMD EPYC 7742 clocked at 2.25 GHz.

### 3.2 EVALUATION METRICS

We use F1 scores to evaluate model performance, it is computed by comparing the imputed labels against a held-out subset of the dataset. We evaluate the model in three steps. First, we compare the average F1 score of our approach against existing imputation baselines to evaluate overall performance (see Table 3 and Table 2). Second, we conduct an ablation study to assess the contribution of each components of the model architecture (see Table 4). Third, we test model's robustness by measuring imputation accuracy under varying levels of missingness (see Tables 5).

## 3.3 RESULTS

### 3.3.1 MODEL ACCURACY

To evaluate imputation accuracy, we hold out 10% of the observed data for each attribute and use them as ground truth for computing F1 scores. Due to lack of observed data we only hold out 10%. Our method is specifically designed to leverage aggregate level expert elicitations, which to best of our knowledge is second of its kind expect for Krapu et al. (2023). Other baselines chosen to compare our method are popular imputation methods and most of them require training data. Following are the baselines our proposed model is compared with:

- **Mode:** Missing values are imputed using the most frequent value in each column.
- **MICE:** Multiple Imputation by Chained Equations van Buuren & Groothuis-Oudshoorn (2011), which iteratively models each variable conditional on the others using regression-based updates.
- **ReMasker:** Transformer-based approach for tabular data imputation Du et al. (2023).
- **OT Imputer:** Optimal transport-based approach for tabular data imputation Muzellec et al. (2020).
- **GBIMC:** Hierarchical Bayesian model with conditional autoregressive priors designed for spatially-informed attribute imputation Krapu et al. (2023).

Table 2: Synthetic data ($N = 5000$) imputation F1 scores.

| Attribute | Missingness | Mode | MICE | OTImputer | GBIMC | EGVAE |
|---|---|---|---|---|---|---|
| **area** (40% missing) | MAR | 0.409 | 0.205 | 0.495 | 0.564 | **0.596** |
| | MCAR | 0.384 | 0.192 | 0.493 | 0.455 | **0.585** |
| | MNAR | 0.414 | 0.207 | 0.482 | 0.430 | **0.637** |
| **height** (100% missing) | MAR | - | - | - | 0.269 | **0.499** |
| | MCAR | - | - | - | 0.121 | **0.424** |
| | MNAR | - | - | - | 0.213 | **0.492** |
| **material** (65% missing) | MAR | 0.427 | 0.237 | 0.511 | 0.524 | **0.669** |
| | MCAR | 0.402 | 0.296 | 0.532 | 0.402 | **0.647** |
| | MNAR | 0.428 | 0.234 | 0.492 | 0.202 | **0.681** |
| **purpose** (90% missing) | MAR | 0.195 | 0.756 | 0.308 | **0.820** | 0.741 |
| | MCAR | 0.415 | 0.451 | 0.594 | 0.595 | **0.671** |
| | MNAR | 0.195 | **0.737** | 0.334 | 0.579 | 0.701 |

| Model | Nashville facade | Nashville roof mat | Nashville purpose | Nashville area | Nashville floors | Amman area | Amman floors | Amman purpose | Mexico City area | Mexico City purpose | Mexico City floors |
|---|---|---|---|---|---|---|---|---|---|---|---|
| Mode* | 0.190 | 0.126 | 0.117 | 0.089 | 0.072 | 0.084 | 0.156 | 0.132 | 0.118 | 0.120 | **0.450** |
| MICE* | 0.190 | 0.126 | 0.117 | 0.089 | 0.072 | 0.084 | 0.156 | 0.132 | 0.118 | 0.120 | **0.450** |
| ReMasker* | **0.382** | 0.051 | 0.168 | **0.415** | 0.103 | - | - | - | - | - | - |
| OTImputer* | 0.237 | 0.174 | 0.160 | 0.305 | 0.120 | 0.150 | 0.217 | 0.132 | 0.132 | 0.120 | 0.307 |
| GBIMC | 0.159 | 0.115 | 0.200 | 0.402 | 0.109 | 0.239 | 0.403 | 0.231 | - | - | - |
| **EGVAE** | 0.289 | **0.176** | **0.178** | 0.391 | **0.121** | **0.220** | **0.318** | **0.235** | **0.171** | **0.325** | 0.324 |

Table 3: F1 Scores across different models for Nashville, Amman and Mexico (* cannot impute columns with no observations).

Table 2 reports the performance of our method and the baselines in terms of F1 score for synthetic data with various types of missingness (MAR: Missing at Random, MCAR: Missing Completely at Random and MNAR: Missing Not at Random) Little & Rubin (2002). EGVAE outperforms other models for most variables across different types of missingness. Additionally, height, which is completely missing in the dataset

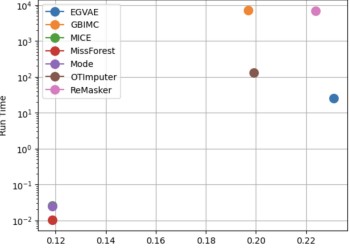

Figure 4: Runtime vs. F1 Score (Nashville data)

imputations can only be generated using GBIMC and EGVAE (− denotes no training data for imputation), where EGVAE markedly outperfoms GBIMC. Table 3 reports the performance for Nashville, Amman and Mexico City building datasets. Results are reported only for attributes with at least some observed data, as most baseline models cannot impute attributes with no training observations. '−' denotes that the process did not complete within the 14-hour time limit. Given the global scalability requirements of this problem, extended runtimes substantially diminish the methodological utility of the approach. Across all read world datasets, we find that naive approaches such as Mode and MICE tend to collapse to imputing the majority class, reflecting the highly skewed nature of the observed data. This bias severely limits their effectiveness for attributes with no observation, heterogeneous or long-tailed distributions. By contrast, our model, which combines learned representations with expert-guided priors, achieves substantially higher F1 scores than the naive models and comparable results to ReMasker and OTImputer when large amount of observed data is available. These results highlight the importance of expert priors for robust imputation in real-world building datasets. GBIMC approach fares better than naive approaches by explicitly modeling spatial correlations and expert priors, but it is computationally intensive and does not scale well to large datasets. Inference requires substantial resources and processing time, which limits its practicality for city-scale applications. In comparison, our model combines learned latent representations from a GNN with expert-guided priors using neural networks to achieve both higher F1 scores and substantially faster runtimes (see Figure 4).

EGVAE, much like other variational autoencoders, is trained with a hybrid loss function. Each of its components directing the model in different direction and building a posterior overtime, such is the case with EGVAE (see Figure 5). During early training, the model often prioritizes reconstructing the input ($\mathcal{L}_{\text{recon}}$ and $\mathcal{L}_{\text{CPT}}$) before it has learned a well-structured latent space ($\mathcal{L}_{\text{VQ}}$) .

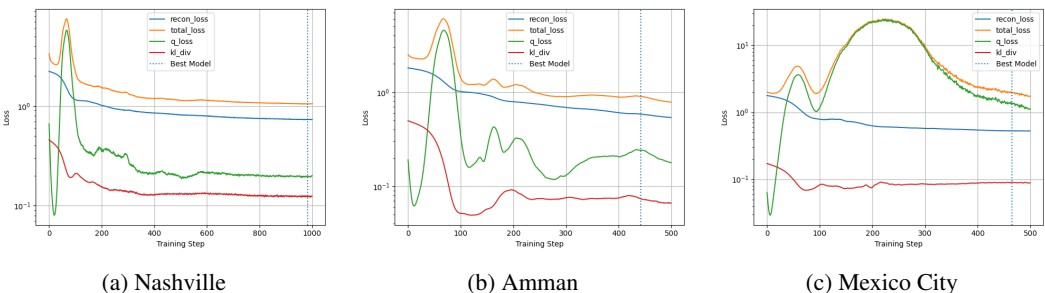

| (a) Nashville | (b) Amman | (c) Mexico City |

Figure 5: EGVAE Training Loss Components

### 3.3.2 ABLATION STUDY

We conduct an ablation study to evaluate the effect of individual components of our model:

- **No CPT Regularization:** To assess the impact of incorporating expert knowledge into our model, we use kl-divergence between imputation with expert priors. We conduct an ablation in which we remove the KL-divergence-based regularization term from the EGVAE loss function. This term guides the model toward expert priors. Without this component, the model retains its ability to perform competitively on attributes with partial observations demonstrating that the data alone can support reasonable inference in these cases (see Table 4). For instance, in Nashville dataset *facade, roof material, purpose* and area have similar KL-divergence value to the full model. This is also the case for partially observed attributes in Amman and Mexico City dataset. However, for attributes with no observed data, the model without expert regularization fails to align its predictions with expert priors. These results underline the importance of expert-guided regularization for improving the plausibility and reliability of imputations in the absence of observed data. Expert regularization aligns imputation to the expert opinion except when observed data is not in line with expert prior such in the case of *floors*.

- **No Graph Structure:** To evaluate the contribution of the graph-based encoder in our model, we perform an ablation study by replacing it with a simple two-layer fully connected feedforward neural network (matching hidden and latent dimensions, $d_h, d_z$ to graph neural network). This version removes any relational inductive bias introduced by the graph structure. The resulting model is outperformed by the full EGVAE model on most attributes (see Table 4). This highlights the value of neighbouring buildings information for improving imputations in the absence of direct observations.

| Variant | Metric | llrs* | mat.* | found.* | shape* | facade | roof mat | purpose | area | floors |
|---|---|---|---|---|---|---|---|---|---|---|
| | | | | | **Nashville** | | | | | |
| EGVAE | F1 Score (↑) | - | - | - | - | **0.289** | **0.176** | **0.178** | **0.391** | **0.121** |
| No GNN | F1 Score (↑) | - | - | - | - | 0.190 | 0.058 | 0.144 | 0.381 | 0.090 |
| EGVAE | KL Div (↓) | **1.502** | **3.376** | **0.400** | **1.215** | **3.595** | **8.126** | **0.391** | **0.462** | 4.263 |
| No Exp. Reg. | KL Div (↓) | 2.008 | 5.022 | 1.018 | 1.746 | 3.680 | 12.296 | **0.391** | 0.478 | **4.093** |

| Variant | Metric | llrs* | mat.* | found.* | shape* | facade* | roof mat* | purpose | area | floors |
|---|---|---|---|---|---|---|---|---|---|---|
| | | | | | **Amman** | | | | | |
| EGVAE | F1 Score (↑) | - | - | - | - | - | - | **0.235** | **0.220** | 0.318 |
| No GNN | F1 Score (↑) | - | - | - | - | - | - | 0.145 | 0.214 | **0.349** |
| EGVAE | KL Div (↓) | **0.607** | **1.409** | **0.001** | **0.127** | **0.455** | **0.087** | 0.043 | 0.177 | 0.431 |
| No Exp. Reg. | KL Div (↓) | 1.313 | 1.868 | 1.092 | 3.864 | 2.500 | 0.447 | **0.041** | **0.171** | **0.423** |

| Variant | Metric | llrs* | mat.* | found.* | shape* | facade* | roof mat* | purpose | area | floors |
|---|---|---|---|---|---|---|---|---|---|---|
| | | | | | **Mexico City** | | | | | |
| EGVAE | F1 Score (↑) | - | - | - | - | - | - | **0.325** | **0.171** | 0.324 |
| No GNN | F1 Score (↑) | - | - | - | - | - | - | 0.124 | 0.149 | **0.381** |
| EGVAE | KL Div (↓) | **0.818** | **0.978** | **0.441** | **0.001** | **1.032** | 0.883 | **0.047** | 0.410 | **0.840** |
| No Exp. Reg. | KL Div (↓) | 1.129 | 1.461 | 1.489 | 1.404 | 1.566 | **0.544** | **0.047** | **0.407** | 1.133 |

Table 4: Ablation study comparing contributions of model components to performance. We report F1 Score (higher is better) and KL-divergence (lower is better) for Nashville and Amman dataset (* no observed data).

### 3.3.3 IMPACT OF MISSINGNESS

To study the robustness of our model to missing data, we vary the missing value rate across three levels: 10%, 20%, and 30%. Table 5 shows relatively stable performance of our model as the missing rate increases. This suggests that the model is able to leverage prior information effectively, offsetting the uncertainty introduced by increased missing values.

| Dataset | Variable | Support | 10% Missing | 20% Missing | 30% Missing |
|---|---|---|---|---|---|
| Nashville | facade | 12 | 0.2898 | 0.3226 | 0.2649 |
| ($N = 18,390$) | roof mat. | 13 | 0.1763 | 0.1501 | 0.1662 |
| | purpose | 9 | 0.1787 | 0.1997 | 0.1979 |
| | area | 6 | 0.3912 | 0.3716 | 0.3675 |
| | floors | 11 | 0.1211 | 0.0863 | 0.0999 |
| Amman | area | 6 | 0.2201 | 0.2116 | 0.2310 |
| ($N = 428,494$) | floors | 4 | 0.3180 | 0.3060 | 0.3054 |
| | purpose | 9 | 0.2352 | 0.2414 | 0.2310 |
| Mexico | area | 6 | 0.1717 | 0.1422 | 0.1750 |
| ($N = 3,194,984$) | floors | 4 | 0.3247 | 0.2996 | 0.2772 |
| | purpose | 9 | 0.3254 | 0.2621 | 0.2599 |

Table 5: Variable support and F1 Scores at various levels of missingness.

### 3.3.4 UMAP OF LATENT REPRESENTATIONS

We project the learned latent representations $Z$ into two dimensions using UMAP McInnes et al. (2020), as shown in Figure 6. In all plots, the points are colored by the attribute *purpose*. The

results demonstrate that our model learns separable latent representations that reflect meaningful distinctions across building types. A similar clustering pattern is observed even when KL-divergence regularization is removed. However, when the graph neural network encoder is replaced with a simple feed-forward neural network, the resulting representations fail to exhibit such clear structure.

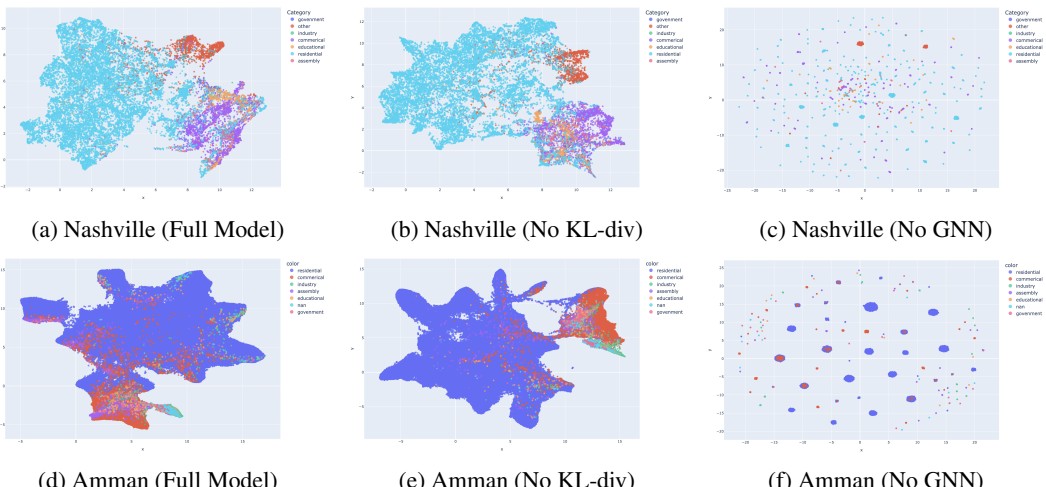

(a) Nashville (Full Model)     (b) Nashville (No KL-div)     (c) Nashville (No GNN)

(d) Amman (Full Model)     (e) Amman (No KL-div)     (f) Amman (No GNN)

Figure 6: UMAP plots for latent representations of buildings.

## 4 CONCLUSION

We introduced Expert-Guided-VAE, leveraging variational-autoencoder framework for modeling tabular data with known structural dependencies and discrete latent representations. By integrating graph-based encoders, vector quantization, and conditional probability table (CPT) regularization, our model effectively incorporates prior knowledge and domain structure into learning. Extensive experiments demonstrate that this combination significantly improves imputation performance, particularly in data-scarce regimes.

Experiments demonstrate that the proposed method outperforms existing methods in terms of imputation accuracy and alignment with expert knowledge (priors). The use of graph-based encoder and CPT regularization significantly improves performance. The ablation studies confirm the critical role of each model component in achieving these results.

Crucially, our approach underscores the value of expert elicitation to define probabilistic dependencies as a key driver of accurate imputation when observational data is limited. By aligning learned representations with expert-informed CPTs, the model ensures semantically consistent and trustworthy predictions, paving the way for reliable deployment in high-stakes applications such as healthcare, finance, and policy modeling.

## 5 REPRODUCABILITY

The source code, along with instructions to reproduce our results for the synthetic data, is available at `https://anonymous.4open.science/r/EGAVE-ICLR-A72E/`. Random seeds are fixed across runs to ensure consistent behavior of stochastic components. Further extension of the mathematical formulation of the model and a pseudo code is fully specified in the Appendix.

## 6 ACKNOLEDGEMENT

We acknowledge the use of ChatGPT for assistance with LaTeX formatting of tables and figures. This work makes use of internal datasets (details provided in the camera-ready). Computational experiments were supported by institutional resources (details provided in the camera-ready).

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

# 7 APPENDIX

This appendix provides the full mathematical specification of the Expert-Guided Variational Autoencoder (EGVAE). We begin by formalizing the observed data and missingness mask, then introduce the two graph structures that underlie the model: the building graph, which captures the spatial or relational similarity between individual buildings, and the attribute directed acyclic graph (DAG), which encodes expert knowledge about dependencies among the building attributes. We then present the encoder architecture based on graph convolutional neural networks (GCNs), the discrete latent representation obtained via vector quantization, and the decoder that produces categorical attribute predictions. Finally we derive the training objective, comprising reconstruction, expert CPT regularization, and VQ losses; as well as the imputation procedure used at test time.

## 7.1 DATA REPRESENTATION AND GRAPH STRUCTURE

Given a categorical tabular dataset, $X$, with $n$ buildings and $d$ building attributes (features), we model the $i^{\text{th}}$ feature of building $j$, denoted $X_{ji}$, as one of $c_i + 1$ categories from $\{-1, 0, 1, \ldots, c_i - 1\}$. The value $X_{ji} = -1$ denotes a missing value while the other $c_i$ categories represent observable quantities of feature $i$. To account for missingness, let $M$ be an $n \times d$ mask matrix, with $M_{ji} = \mathbb{I}\{X_{ji} \neq -1\}$, where $\mathbb{I}$ is an indicator function. Under the assumption that buildings maintain similar physical properties, dictated by as building codes, and spatial properties, dictated by zoning and administrative boundaries, we propose two graph structures.

The spatial properties are identified through a building graph, $\mathcal{G} = (B, P)$, where the nodes are buildings such that $|B| = n$ and $P$ is the set of all edges, determined by an adjacency matrix, $A$. For two buildings indexed by $u$ and $v$, the value $A_{uv} = A_{vu}$ is defined as $A_{uv} = \mathbb{I}\{\|\text{loc}(u) - \text{loc}(v)\|_2 \leq r\}$, where $\text{loc}(u)$ is the spatial coordinate of building $u$.

The physical properties are identified through a directed acyclic graph, $\mathcal{D} = (V, E)$, where $V = \{1, 2, \ldots, d\}$ represents the vertices of the graph being the $d$ features and $E$ are a collection of conditional relationships between the $d$ features. The structure of $\mathcal{D}$ induces a Parent-Child relationship between the features and can be captured through a topological ordering of the graph defined by

$$\text{Pa}(i) = \{p \in V : B_{pi} = 1\}, \qquad p \prec i \quad \forall p \in \text{Pa}(i)$$

and $B_{pi} = \mathbb{I}\{(p \to i) \in E\}$. Here, $B$ is a $d \times d$ Parent-Child mask. Then, given $\mathcal{D}$, for each attribute $i$, experts provide a conditional categorical

$$\pi_i\left(\cdot | x_{\text{Pa}(i)}\right) \in \Delta^{c_i - 1}, \quad \pi(x_1, \ldots, x_d) = \prod_{i=1}^{d} \pi_i\left(x_i | x_{\text{Pa}(i)}\right),$$

where the values of $x$ are the categorical attribute assignments and $\Delta$ is the probability simplex

$$\Delta_{c_i - 1} = \left\{ p \in \mathbb{R}^{c_i} : p_l \geq 0, \sum_{l=1}^{c_i} p_l = 1 \right\}.$$

Hence the quantity $\pi_i\left(\cdot | x_{\text{Pa}(i)}\right)$ is a conditional distribution for attribute $i$. To obtain the target distribution, $\tilde{x}$, let

$$\prod_i \in \mathbb{R}^{c_i \times \prod_{p \in \text{Pa}(i)} c_p}$$

collect $\pi_i$ columns by parent configuration. Using one-hot parent vectors $e_{x_p}^{(p)} \in \{0, 1\}^{c_p}$ and a Kronecker selector

$$\eta_{j, \text{Pa}(i)} = \bigotimes_{p \in \text{Pa}(i)} e_{\tilde{x}_{jp}}^{(p)} \in \{0, 1\}^{\prod_{p \in \text{Pa}(i)} c_p}.$$

The expert target distribution for attribute $j$ is then

$$\pi_i\left(\cdot|\tilde{x}_{j,\text{Pa}(i)}\right) = \prod_i \eta_{j,\text{Pa}(i)} \in \Delta^{c_i-1}.$$

In the case of missing data, such that $\boldsymbol{M}_{jp} = 0$ for $p \in \text{Pa}(i)$, we see that $\eta_{j,\text{Pa}(i)}$ is undefined as we will not know which CPT column to to select. To resolve this, we will need a plug-in rule for missing parents; to which the decoder's categorical predictions, $q_{\theta,p}(\cdot|\hat{z}_j)$, can be used in a mean-field MAP value. Construction of $q_{\theta,p}(\cdot|\hat{z}_j)$ will be developed during discussion of the decoder.The resulting plug-in value is then

$$\tilde{x}_{jp} = \begin{cases} X_{jp}, & M_{jp} = 1 \quad \text{(observed)}, \\ \\ \underset{\ell \in \{0,\ldots,c_p-1\}}{\arg\max} q_{\theta,p}(\ell \mid \hat{z}_j), & M_{jp} = 0 \quad \text{(missing)}. \end{cases}$$

The CPT component, $\mathcal{L}_{\text{CPT}}$, of the global loss function, $\mathcal{L}_{\text{total}}$, is then given by

$$\mathcal{L}_{\text{CPT}} = \frac{1}{n} \sum_{j=1}^{n} \sum_{i=1}^{d} \text{KL}\big(q_{\theta,i}(\cdot \mid \hat{z}_j) \,\big\|\, \pi_i(\cdot \mid \tilde{x}_{j,\text{Pa}(i)})\big).$$

## 7.2 Vector Quantized Autoencoder

In all entries where the mask matrix, $\boldsymbol{M}$ is zero, an imputation process is needed to complete the data matrix, $\boldsymbol{X}$. Here, we propose a Vector Quantized Autoencoder that operates in three stages: encode, vector quantize, decode. The encoder process takes the observed data, $\boldsymbol{X}$, and propagates information across the building graph, $\mathcal{G}$. The vector quantization process compresses each building's representation into a discrete latent code, $\hat{z}_j$. The decoder process uses $\hat{z}_j$ to predict all attributes of the building, thereby creating a categorial distribution, $q_{\theta,i}(\cdot|\hat{z}_j)$.

### 7.2.1 Encode Process

For the encode process, each feature $\boldsymbol{X}_i$ is embedded using a learnable embedding matrix, $\boldsymbol{E}_i \in \mathbb{R}^{(c_i+1)+d_e}$, where the last row handles the missing value indicator. Then, for building $j$, the initial feature vector is the concatenation of all feature embeddings

$$h_j^{(0)} = \left[E_1^T \xi_{j1} || E_2^T \xi_{j2} || \ldots || E_d^T \xi_{jd}\right],$$

where $\xi_{ji}$ is the one-hot encoding for feature $i$. Collecting all rows, letting $d_z = d \times d_e$ we obtain $\boldsymbol{H} \in \mathbb{R}^{n \times d_z}$, which we denote the full input embedding. To encode these embeddings, we appeal to the Kipf-Welling Graph Convolutional Network (GCN) using the spatial adjacency matrix, $\boldsymbol{A}$, from the building graph $\mathcal{G}$ and the full input embedding. Under this construction, the first layer of the encoding stage is obtained through a ReLU activation of a composition of the normalized adjacency graph and the full input embedding as

$$\max\left\{0, \left(\text{diag}\left((\boldsymbol{A}+\boldsymbol{I})\boldsymbol{J}\right)\right)^{-\frac{1}{2}} (\boldsymbol{A}+\boldsymbol{I}) \left(\text{diag}\left((\boldsymbol{A}+\boldsymbol{I})\boldsymbol{J}\right)\right)^{-\frac{1}{2}} \boldsymbol{H}\boldsymbol{W}\right\},$$

where $\boldsymbol{J}$ is a vector of all one and $\boldsymbol{W}$ is a weight matrix. Note that $\text{diag}\left((\boldsymbol{A}+\boldsymbol{I})\boldsymbol{J}\right)$ is the degree matrix. Next, at each stage of the GCN, we impose a $\alpha\%$ dropout using a random Bernoulli mask matrix, $\boldsymbol{M}_\alpha$, where $[\boldsymbol{M}_\alpha]_{ji} = 1$ with probability $(1-\alpha)$. The resulting first layer of the three-layer GCN is given by

$$\boldsymbol{Z}^{(1)} = \boldsymbol{M}_\alpha^{(0)} \odot \max\left\{0, \left(\text{diag}\left((\boldsymbol{A}+\boldsymbol{I})\boldsymbol{J}\right)\right)^{-\frac{1}{2}} (\boldsymbol{A}+\boldsymbol{I}) \left(\text{diag}\left((\boldsymbol{A}+\boldsymbol{I})\boldsymbol{J}\right)\right)^{-\frac{1}{2}} \boldsymbol{H}^{(0)}\right\}.$$

Here, we introduce a superscript to identify the input and output of each layer within the GCN. For all three layers, we obtain

$$\boldsymbol{Z}^{(1)} = \boldsymbol{M}_\alpha^{(0)} \odot \max\left\{0, \left(\mathrm{diag}\left((\boldsymbol{A}+\boldsymbol{I})\,\boldsymbol{J}\right)\right)^{-\frac{1}{2}} (\boldsymbol{A}+\boldsymbol{I}) \left(\mathrm{diag}\left((\boldsymbol{A}+\boldsymbol{I})\,\boldsymbol{J}\right)\right)^{-\frac{1}{2}} \boldsymbol{H}^{(0)}\boldsymbol{W}^{(0)}\right\}, \tag{8}$$

$$\boldsymbol{Z}^{(2)} = \boldsymbol{M}_\alpha^{(1)} \odot \max\left\{0, \left(\mathrm{diag}\left((\boldsymbol{A}+\boldsymbol{I})\,\boldsymbol{J}\right)\right)^{-\frac{1}{2}} (\boldsymbol{A}+\boldsymbol{I}) \left(\mathrm{diag}\left((\boldsymbol{A}+\boldsymbol{I})\,\boldsymbol{J}\right)\right)^{-\frac{1}{2}} \boldsymbol{Z}^{(1)}\boldsymbol{W}^{(1)}\right\}, \tag{9}$$

$$\boldsymbol{Z} = \left(\mathrm{diag}\left((\boldsymbol{A}+\boldsymbol{I})\,\boldsymbol{J}\right)\right)^{-\frac{1}{2}} (\boldsymbol{A}+\boldsymbol{I}) \left(\mathrm{diag}\left((\boldsymbol{A}+\boldsymbol{I})\,\boldsymbol{J}\right)\right)^{-\frac{1}{2}} \boldsymbol{Z}^{(2)}\boldsymbol{W}^{(2)}. \tag{10}$$

$$\tag{11}$$

A single forward pass will yield $\boldsymbol{Z}$ with parameter set $\phi = \left\{E_i, W^{(0)}, W^{(1)}, W^{(2)}\right\}$, dropout masks $\omega = \left\{\boldsymbol{M}^{(0)}, \boldsymbol{M}^{(1)}\right\}$, and inputs $\boldsymbol{A}$ and $\boldsymbol{H}^{(0)}$. As a result, $\boldsymbol{Z}$ is a collection of latent vectors, $z_j$, each of length $d_z$. Noting that index $j$ corresponds to building $j$, we call $z_j$ a latent code for building $j$.

### 7.2.2 VECTOR QUANTIZATION PROCESS

Given these latent codes, we then define a codebook of $K$ embeddings, $\mathcal{E} = \{e_1, e_2, \ldots, e_K\}$ with $e_k \in \mathbb{R}^{d_z}$. The vector quantization process assigns each latent code to its nearest codebook vector, defined via

$$\hat{z}_j = e_{\ell^*(j)}, \qquad \text{where} \quad \ell^*(j) = \underset{\ell \in \{1,2,\ldots,K\}}{\arg\min} \; ||z_j - e_\ell||_2^2.$$

Through assignment of the latent codes to the codebook vectors, some amount of error exists, which requires training during optimization. To this end, a forward pass optimization is used from the encode stage through the vector quantization stage and through the decode stage. After the forward pass is completed, a backward pass optimization is used. At the vector quantization stage, these optimization passes are handled differently. To control for forward and backward passes, the stop-gradient operator, $\mathrm{sg}[\cdot]$, is applied. Here, $\mathrm{sg}[z_j] = z_j * \mathbb{I}\{\text{forwardpass}\}$. This means that during the forward pass, we obtain the expression $\hat{z}_j = e_{\ell^*(j)}$. However in the backward pass, the gradient with respect to $z_j$ passes through the $z_j$ term only, while the gradient with respect to the codebook vector $e_{\ell^*(j)}$ passes through $\left(e_{\ell^*(j)} - \mathrm{sg}[z_j]\right)$. This prevents double-counting in the gradient in the backward pass. Thus, we obtain the vector quantization component of the loss function, $\mathcal{L}_{\mathrm{VQ}}$, of the global loss function, $\mathcal{L}_{\mathrm{total}}$, given by

$$\mathcal{L}_{\mathrm{VQ}} = \frac{1}{n} \sum_{j=1}^n \left(\|\mathrm{sg}[z_j] - \hat{z}_j\|_2^2 + \beta \, \|z_j - \mathrm{sg}[\hat{z}_j]\|_2^2\right),$$

where $\beta > 0$ balances encoder commitment against codebook stability. The result is a vector quantized, discrete latent code $\hat{Z} = [\hat{z}_1; \hat{z}_2; \ldots; \hat{z}_n]$. Furthermore, the learnable parameters through $\mathcal{L}_{\mathrm{VQ}}$ are the encoder parameters, $\phi$ through the commitment term, $\beta \, \|z_j - \mathrm{sg}[\hat{z}_j]\|_2^2$, and the codebook embeddings, $\{e_k\}$ through the codebook update term, $\|\mathrm{sg}[z_j] - \hat{z}_j\|_2^2$.

### 7.2.3 DECODE PROCESS

In the decode process, the quantized latent representations, $\hat{Z}$, are mapped into categorical probability distributions over the attribute categories. Specifically, each quantized vector $\hat{z}_j$ is first transformed through a hidden layer:

$$h_j = \mathrm{ReLU}(W^{(h)}\hat{z}_j + b^{(h)}), \quad h_j \in \mathbb{R}^{d_h},$$

where $W^{(h)} \in \mathbb{R}^{d_h \times d_z}$ and $b^{(h)} \in \mathbb{R}^{d_h}$. Then, for each feature $i \in \{1, 2, \ldots, d\}$ with $c_i$ categories, a softmax projection produces the predicted categorical distribution:

$$q_{\theta,i}(\ell \mid \hat{z}_j) = \frac{\exp\left((W^{(i)}h_j + b^{(i)})_\ell\right)}{\sum_{m=0}^{c_i-1} \exp\left((W^{(i)}h_j + b^{(i)})_m\right)}, \quad \ell = 0, \ldots, c_i - 1,$$

where $W^{(i)} \in \mathbb{R}^{c_i \times d_h}$ and $b^{(i)} \in \mathbb{R}^{c_i}$. Then $q_{\theta,i}(\cdot \mid \hat{z}_j) \in \Delta^{c_i-1}$ is the decoder's predicted probability distribution for feature $i$. From this model, the probability of the observed feature for building $j$ is given by

$$p_\theta(x_{j\cdot} \mid \hat{z}_j) = \prod_{i=1}^{d} \prod_{\ell=0}^{c_i-1} q_{\theta,i}(\ell \mid \hat{z}_j)^{\mathbb{I}\{x_{ji}=\ell\}\boldsymbol{M}_{ji}},$$

where $\boldsymbol{M}_{ji}$ is the missingness mask matrix and $\mathbb{I}\{x_{ji} = \ell\}$ is the indicator of the true category. Then, the reconstruction component of loss function, $\mathcal{L}_{\mathrm{recon}}$, of the global loss function, $\mathcal{L}_{\mathrm{total}}$, is given by

$$\mathcal{L}_{\mathrm{recon}}(\theta) = -\frac{1}{n} \sum_{j=1}^{n} \sum_{i=1}^{d} M_{ji} \log q_{\theta,i}(x_{ji} \mid \hat{z}_j).$$

The parameters for the reconstruction loss are the decoder parameters, the encoder parameters, $\phi$, and the codebook parameters, $\mathcal{E}$. The decoder parameters are given by

$$\theta = \left\{ W^{(h)}, b^{(h)}, W^{(1)}, b^{(1)}, W^{(2)}, b^{(2)}, \ldots, W^{(d)}, b^{(d)} \right\},$$

which are the weights and biases for the hidden layer and the respective $d$ features. It should be noted that the weights here are not the same as in the encoder process.

---

**Algorithm 1:** Train EGVAE

---

**Input:** Graph data $(x, \texttt{edge\_index})$, hyperparameters
**Output:** Trained EGVAE parameters
$z \leftarrow \texttt{GraphEncoder}(x, \textit{edge\_index})$;
$(\hat{z}, \ell_{\mathrm{VQ}}) \leftarrow \texttt{VectorQuantizer}(z)$;
$\hat{x} \leftarrow \texttt{Decoder}(\hat{z})$;
$\ell_{\mathrm{recon}} = \texttt{Cross-Entropy}(\hat{x}_{obs}, x_{obs})$;
$\ell_{\mathrm{KL}} = \texttt{KL-Div}(\hat{x}, x)$;
Total loss: $\ell = \ell_{\mathrm{recon}} + \ell_{\mathrm{VQ}} + \ell_{\mathrm{KL}}$;
Update model parameters with gradient step on $\ell$;

---

