# OpenReview forum: "Imputing Incomplete Building Attribute Data Using Expert-Guided Variational Autoencoders (EGVAE)"
_ICLR.cc/2026/Conference — Submitted to ICLR 2026_

### Official Review · Reviewer_aCgK · 2025-10-31

**Soundness:** 3
**Presentation:** 2
**Contribution:** 3
**Rating:** 6
**Confidence:** 2

**Summary:**

Although infrastructure attribute data are important,
regions with scarce data suffer from many missing values.
This paper proposes Expert-Guided Variational Autoencoders (EGVAE),
a VQ-VAE with a GNN encoder that captures spatial correlations and incorporates expert prior knowledge via a KL loss.
The proposed method can learn latent variables that enable imputation even for attributes with little to no observed data.
Experiments on various real-world datasets show that the proposed method outperforms existing approaches, especially in data-scarce settings.

**Strengths:**

I am not an expert in this field, but the proposed method,
which combines a VQ-VAE with a GNN encoder and regularization based on conditional probability tables,
appears simple yet effective.
This method achieves strong performance on real data,
and the ablation study sufficiently demonstrates which modules are effective.

**Weaknesses:**

Please see the Questions section.

**Questions:**

- In Section 2.3, the introduced loss function does not appear to weight the individual loss terms. I suspect that adding hyperparameters as weights to each term could improve performance. Did you conduct experiments along these lines?
- For missing-data imputation, I believe CPT-based regularization is important, yet in Figure 6 the No KL-div condition looks quite similar. What effect does this regularization have in the latent variable space?

---

### Official Review · Reviewer_mqzK · 2025-10-31

**Soundness:** 2
**Presentation:** 3
**Contribution:** 2
**Rating:** 4
**Confidence:** 4

**Summary:**

The authors propose EGVAE (Expert-Guided Variational Autoencoder) to impute incomplete building attribute data (e.g., roof material, number of floors) in large urban datasets. The model combines a graph neural encoder to capture inter-attribute and inter-sample relationships, a vector-quantization module in latent space, and an expert-guided prior via conditional probability tables in a KL-regularization term to incorporate domain knowledge. Experiments on datasets from multiple cities show improved F1 scores and robustness under high missingness compared to traditional imputation baselines.

**Strengths:**

The paper is clearly written and well-structured. The proposed integration of a graph neural encoder, vector-quantized latent space, and expert-guided prior is technically sound for the task of imputing missing building attributes. Experimental results across multiple cities also demonstrate the effectiveness.

**Weaknesses:**

- Novelty. The overall architecture of EGVAE combines several well-established components rather than introducing a new mechanism. While the integration of a graph encoder, vector quantization, and expert-guided prior is reasonable, it feels incremental.

- Problem scope. The problem of missing building attributes is too domain-specific and kind of narrow for an ICLR-level contribution. The current status feels more like an application-specific contribution rather than a broadly applicable framework for imputation. Even though spatial adjacency is used in the method, I believe the method can also apply to a more general missing-data problems (e.g., healthcare, finance, or tabular data).

- Missing related work.
The paper compares only against general-purpose imputation methods, overlooking VAE-based imputation models [1, 2, 3] that directly address missing data or tabular data. It is important to discuss their connections.

   [1] Alfredo Nazabal et al., Handling Incomplete Heterogeneous Data using VAEs

   [2] Yu Gong et al., Variational Selective Autoencoder: Learning from Partially-Observed Heterogeneous Data, AISTATS 2021

   [3] Tom Joy et al., Learning Multimodal VAEs through Mutual Supervision, ICLR 2022


- Unfair comparison. EGVAE leverages spatial adjacency information through a graph built from building proximity, while other baselines do not see such relational structure. Since nearby buildings naturally share similar attributes, this spatial prior provides EGVAE with an inherent advantage.

- Experiment.
(1) The authors should include error bars to indicate the statistical significance of the reported results;
(2) In Table 3 (Amman), the best F1 scores are incorrectly highlighted. GBIMC outperforms EGVAE on two of the three attributes;
(3) On the Mexico City dataset, the "floors" attribute has the highest missing ratio, yet EGVAE performs worse than several baselines. It seems like that the model handles well-observed attributes more effectively than heavily missing ones.

**Questions:**

- Do the authors try incorporating the graph structure (currently used in the CPT prior) into the encoder? If spatial relations are meaningful for the prior, they should also improve posterior inference.
- Minor points: (1) Figure 4 is not correctly formatted; (2) Line 30 "there"-> "their"

---

### Official Review · Reviewer_bycB · 2025-11-01

**Soundness:** 1
**Presentation:** 1
**Contribution:** 1
**Rating:** 2
**Confidence:** 4

**Summary:**

This paper addresses the problem of missing building attribute data in spatial datasets, proposing a deep learning method that combines spatial structure, limited observations, and expert-informed priors. The model is built upon a Vector Quantized Variational Autoencoder (VQ-VAE) with a Graph Neural Network (GNN) encoder to capture spatial dependencies among buildings. A KL-divergence-based regularization term is introduced to integrate prior knowledge. The goal is to enable robust imputation even in data-scarce settings. Experimental results on real-world datasets are presented to support the model's effectiveness.

**Strengths:**

- The application domain is relevant and practically important.

- The use of real-world datasets adds practical relevance and demonstrates the applicability of the method.

**Weaknesses:**

- **Limited novelty**: The core techniques—GNNs, VQ-VAEs, and auxiliary losses via KL regularization—are well-established and widely studied. For example, [1] offers a foundational treatment of variational graph autoencoders. The combination here does not offer a substantial methodological advance.

- **Presentation issues**: The methodological section lacks sufficient detail and justification. Many architectural choices are simply listed without accompanying analysis or rationale, and gaps in the narrative hinder readability. Additionally, formatting issues—such as the blank space on page 7—detract from the professionalism of the presentation. Overall, the paper would benefit from a clearer and more structured exposition.

- **Weak experimental evaluation**: The baselines used for comparison are outdated, and many recent methods for deep generative imputation are neither cited nor evaluated. Numerous VAE-based approaches could be directly tested with the proposed encoder (e.g., [2, 3, 5, 6]).

- **Unjustified design choices**: The motivation for using a quantized latent space (via VQ-VAE) over a continuous latent space is not discussed. It remains unclear how the model would perform with a standard VAE and whether the quantization adds significant value.

- **Insufficient related work coverage**: The related work section does not reflect the current state of the art. Many recent approaches to imputation, especially those based on deep generative models or diffusion models, are omitted.

- **Reference imprecision**: Some references are incorrectly used. For instance, line 046 cites Nazabal et al. (2018) [2] in the context of VQ-VAEs, but that work uses a standard VAE, not a quantized latent space.

### Typos and Minor Issues

- Line 476: *Reproducability* → *Reproducibility*

- Line 481: *Acknoledgement* → *Acknowledgement*

### References
[1] Kipf, T. N., & Welling, M. (2016). Variational graph auto-encoders. arXiv preprint arXiv:161Kipf, T. N., & Welling, M. (2016). Variational graph auto-encoders. arXiv preprint arXiv:161

[2] Nazabal, Alfredo, et al. "Handling incomplete heterogeneous data using vaes." Pattern Recognition 107 (2020): 107501.

[3] Ma, Chao, et al. "Vaem: a deep generative model for heterogeneous mixed type data." Advances in Neural Information Processing Systems 33 (2020): 11237-11247.

[4] Yoon, Jinsung, James Jordon, and Mihaela Schaar. "Gain: Missing data imputation using generative adversarial nets." International conference on machine learning. PMLR, 2018.

[5] Peis, Ignacio, Chao Ma, and José Miguel Hernández-Lobato. "Missing data imputation and acquisition with deep hierarchical models and hamiltonian monte carlo." Advances in Neural Information Processing Systems 35 (2022): 35839-35851.

[6] Mattei, Pierre-Alexandre, and Jes Frellsen. "MIWAE: Deep generative modelling and imputation of incomplete data sets." International conference on machine learning. PMLR, 2019.

[7] Jarrett, Daniel, et al. "Hyperimpute: Generalized iterative imputation with automatic model selection." International Conference on Machine Learning. PMLR, 2022.

[8] Zheng, Shuhan, and Nontawat Charoenphakdee. "Diffusion models for missing value imputation in tabular data." arXiv preprint arXiv:2210.17128 (2022).

**Questions:**

I have no further inquiries beyond the issues outlined above.

---

### Meta-Review · Area_Chair_eCWj · 2026-01-09

**Summary:**

The reviewers raised several and important concerns regarding the novelty of the paper, incompleteness of related work, weak empirical evaluation, etc. Unfortunately, no rebuttal has been submitted by the authors and thus those concerns remain.

**Reviewer Concerns:**

All concerns remain.

**Reviewer Scores:**

I do not expect an increase in score due to the lack of a rebuttal.

---

### Decision · Program_Chairs · 2026-01-26

Reject